# DCFNet: Infrared and Visible Image Fusion Network Based on Discrete Wavelet Transform and Convolutional Neural Network

**DOI:** 10.3390/s24134065

**Published:** 2024-06-22

**Authors:** Dan Wu, Yanzhi Wang, Haoran Wang, Fei Wang, Guowang Gao

**Affiliations:** School of Electronic Engineering, Xi’an Shiyou University, Xi’an 710312, China; wangyanzhi780@gmail.com (Y.W.); 21212030401@stumail.xsyu.edu.cn (H.W.); 200102@xysu.edu.cn (F.W.); wwgao@xsyu.edu.cn (G.G.)

**Keywords:** image fusion, convolutional neural network, discrete wavelet transform, infrared and visible images

## Abstract

Aiming to address the issues of missing detailed information, the blurring of significant target information, and poor visual effects in current image fusion algorithms, this paper proposes an infrared and visible-light image fusion algorithm based on discrete wavelet transform and convolutional neural networks. Our backbone network is an autoencoder. A DWT layer is embedded in the encoder to optimize frequency-domain feature extraction and prevent information loss, and a bottleneck residual block and a coordinate attention mechanism are introduced to enhance the ability to capture and characterize the low- and high-frequency feature information; an IDWT layer is embedded in the decoder to achieve the feature reconstruction of the fused frequencies; the fusion strategy adopts the $l_{1}-norm$ fusion strategy to integrate the encoder’s output frequency mapping features; a weighted loss containing pixel loss, gradient loss, and structural loss is constructed for optimizing network training. DWT decomposes the image into sub-bands at different scales, including low-frequency sub-bands and high-frequency sub-bands. The low-frequency sub-bands contain the structural information of the image, which corresponds to the important target information, while the high-frequency sub-bands contain the detail information, such as edge and texture information. Through IDWT, the low-frequency sub-bands that contain important target information are synthesized with the high-frequency sub-bands that enhance the details, ensuring that the important target information and texture details are clearly visible in the reconstructed image. The whole process is able to reconstruct the information of different frequency sub-bands back into the image non-destructively, so that the fused image appears natural and harmonious visually. Experimental results on public datasets show that the fusion algorithm performs well according to both subjective and objective evaluation criteria and that the fused image is clearer and contains more scene information, which verifies the effectiveness of the algorithm, and the results of the generalization experiments also show that our network has good generalization ability.

## 1. Introduction

Image fusion techniques aim at fusing two or more source images from different imaging sensors for the same scene to be able to obtain fused images that are rich in information and high in clarity [1]. Visible-light images have high resolution and contain detailed textural information about the scene [2]; however, under poor external conditions, their contrast diminishes, and their dynamic range becomes insufficient, whereas infrared images utilize thermal radiation information to capture hidden targets even in complex environments, with better contrast and dynamic range but with lower image resolution [3]. By fusing the two through infrared and visible-light image fusion technology, complementary information can be obtained, and fused images with more comprehensive and accurate scene descriptions can be generated. As a result, infrared–visible fusion has a wide range of applications in areas such as military combat, healthcare, and industrial inspection.

In general, infrared and visible image fusion algorithms can be broadly categorized into two groups: traditional methods and deep learning-based methods [4]. Traditional methods mainly include subspace-based [5], sparse representation-based [6], saliency-based [7], and multi-scale transformation-based [8] methods. Subspace-based methods mainly include methods such as principal component analysis (PCA) [9] and independent component analysis (ICA) [10]. Subspace is utilized as a projection of a high-dimensional input image into a low-dimensional space, where the low-dimensional subspace captures the intrinsic structure of a natural image with multiple pieces of redundant information. Methods based on sparse representations [11] suffer from two key problems: sparse coefficient solving and overcomplete dictionary construction. This type of method represents the source image by a small number of linear vectors of the dictionary, measures the activity of the vectors, and completes the weight assignment; finally, it reconstructs the fused image by inverse transformation. Although the sparse representation-based method improves the contrast, it is prone to lose the structural information of the source image, and there is the problem of fuzzy extraction of detailed texture. The saliency-based approach extracts the salient regions in an image by enhancing the important regions in the image and suppressing the background regions [12]. Traditional MST-based image fusion methods include those based on pyramid [13], wavelet transform [14], and multi-scale geometric analysis [15], where the multi-scale transform method usually includes three steps. First, the source image is decomposed to obtain the corresponding low- and high-frequency information. Second, low-frequency information and high-frequency information are fused separately according to the designed fusion rules. Finally, the fused low-frequency fusion information and high-frequency fusion information are inverted to obtain the final fused image.

The multi-scale transform image fusion method can capture the feature information of different scales of the source image and solve the problem of uneven distribution of feature information. However, when the image is transformed at multiple scales, discontinuities or artifacts may be introduced at the edges; on the other hand, the fusion process relies too much on manually set fusion rules, with high computational complexity, and the fusion efficiency is correspondingly difficult to improve. Deep learning-based methods have achieved remarkable research results in the field of image fusion in recent years, which have greatly promoted the technological development in this field. These methods extract the feature maps of the input source images, splices the feature information of the feature maps, and reconstructs them to obtain the final fused image. Compared with traditional fusion algorithms, deep learning-based fusion algorithms do not need to manually define complex rules and processes, have powerful feature learning and generalization capabilities, and are able to automatically perform feature extraction and image reconstruction. They adopt an end-to-end learning approach to effectively capture the global information of the image, avoiding the problems of information loss and local optimization in traditional methods, thus achieving better fusion results.

Therefore, an image fusion algorithm based on discrete wavelet transform and convolutional neural network is proposed by combining the respective advantages of multi-scale transform (MST) and deep learning (DL). In image processing, the source image can be decomposed into different frequency features by using discrete wavelet transform (DWT). The high-frequency sub-bands retain the information of edges and textures, while the low-frequency sub-bands retain the global information in the source image. Then, the extracted high-frequency features and low-frequency features are processed by using a high-frequency detailed feature branching network and a low-frequency global feature branching network, respectively, after which the low-frequency feature information and high-frequency feature information are fused by adopting the designed fusion strategy. The IDWT module is introduced in the decoder, and the fused low-frequency information and high-frequency information are utilized to realize feature reconstruction by discrete wavelet inverse transform; finally, the fused image is obtained by dimensionality reduction in the convolutional layer. This method can effectively solve the problems of activity level measurement design and weight assignment strategy and solve the tricky artifact and boundary effects, and the quality of the obtained fused images is significantly improved. Experimental results for different image fusion scenarios show that our method achieves better performance in both subjective and objective evaluations compared with other state-of-the-art methods.

The main contributions of this paper can be summarized as follows:A DWT-CNN fusion network framework (DCF) is designed for extracting and fusing low-frequency global features and high-frequency detailed features. DWT and IDWT layers are introduced in the designed network to optimize the processing for different frequency-domain feature information. The encoder is embedded in the DWT layer, and the key information of the feature map can be fully utilized to extract the low-frequency and high-frequency feature information of the image through multi-scale decomposition. Among them, the low-frequency feature information corresponds to the smooth region of the image, and the high-frequency feature information corresponds to the edge or texture region. In addition, the feature extraction branching channel of the coding network improves the capture of frequency feature information. The decoder, on the other hand, is embedded in the IDWT layer, and frequency feature reconstruction is realized by discrete wavelet inversion operation to ensure the integrity of the image and accurate representation of the details; the reconstructed image is obtained from the final decoder output.The residual module is introduced into the designed coding network to enhance the feature extraction of low-frequency and high-frequency information and maximize the retention of global and detailed information of different frequency features. Meanwhile, in order to suppress the influence of redundant information on the feature extraction operation, this paper introduces the coordinate attention mechanism, which significantly enhances the network’s ability to capture key information by dynamically adjusting the spatial dependencies in the feature map. In addition, by assigning different weights to each spatial location of the input features, the network model’s response to important features is strengthened, which improves the characterization effect of the features and the overall network performance.

The rest of the paper is organized as follows: Section 2 briefly describes the work related to image fusion algorithms based on multi-scale transforms and deep learning-based image fusion algorithms. In Section 3, the details of the proposed infrared and visible image fusion method are presented. Section 4 shows the related experiments, and conclusions are given in Section 5.

## 2. Related Work

This section briefly reviews traditional multi-scale-based image fusion methods as well as deep learning-based image fusion methods.

### 2.1. Image Fusion Algorithms Based on Traditional Multi-Scale Transforms

MST-based methods include multi-scale decomposition, multi-scale fusion, and multi-scale reconstruction steps. Burt et al. first proposed the Laplace pyramid image fusion method [16]. Ranchin et al. applied the discrete wavelet transform (DWT) method to the field of image fusion and achieved better results [17]. Li proposed a fusion method that combines NSST with bootstrap filtering [18] to reduce the loss of information such as edges and details in the fused image. Zhou et al. [19] proposed a hybrid multi-scale decomposition (hybrid-MSD) image fusion method using multi-scale Gaussian filters and bilateral filters to improve the perceptual information. Yin et al. [20] proposed a novel image fusion algorithm (SIDCST) based on translation-invariant dual-tree complex shear wave transform and sparse representation, and for low-frequency sub-band coefficients, an SR-based fusion rule was proposed. For high-frequency sub-band coefficients, a fusion rule based on adaptive two-channel pulse-coupled neural network is proposed, and the fusion effect is significantly improved. H. Hao et al. [21] designed an infrared and visible image fusion algorithm based on multi-scale decomposition optimization and gradient-weighted local energy (MGFuse), which effectively preserves the richness of the details and sufficiently optimizes and improves the fusion quality. Singh et al. [22], in their paper, proposed a framework for medical image fusion based on discrete fractional wavelets and non-subsampled directional filter banks. Their method utilizes the discrete fractional wavelet transform to perform multi-scale decomposition of medical images and fuses the decomposed low- and high-frequency sub-bands by a non-subsampled directional filter bank. This method can effectively preserve the structure and details at different scales and fuses these sub-bands with weights by using perceptual weights. This method is able to improve the perceptual quality and visual comfort of the fused image while maintaining the image detail information of medical images and improve the contrast and clarity of the images. On the other hand, Paul Hil et al. [23] explored the use of wavelets for perceptual image fusion in their study. Their method is based on a perceptual model that optimizes the perceptual quality of an image by decomposing the image into wavelets. In addition, Haonan Su et al. [24] proposed a multispectral fusion and denoising method for visible and near-infrared images using multi-scale wavelet analysis. Their method achieves multispectral image generation by performing wavelet decomposition of visible and near-infrared images and weighted fusion by combining the multi-scale decomposition coefficients. Also, they utilize a wavelet domain denoising technique for noise reduction of the fused images to improve image quality and detail retention. In summary, DWT-based image fusion methods have a wide range of applications in the fields of medical images, perceptual images, and multispectral images. By utilizing the multi-scale decomposition and reconstruction capabilities of DWT, these methods achieve the fusion and enhancement of structural and detail information of images, which improves the quality and visual effect of images. These studies provide useful references and explorations for DWT-based image fusion, enriching the research contents and methods in the field of image fusion.

However, there are some obvious drawbacks and limitations of traditional multi-scale-based infrared and visible image fusion algorithms. First, the loss or confusion of detail information in the infrared and visible images can occur during the image fusion process, resulting in the fused image not being able to accurately retain the detail nuances of the original image, which reduces the quality and clarity of the image. Second, traditional multi-scale image fusion algorithms rely too much on artificial rules for image fusion, which may not be able to adapt or provide satisfactory fusion results when dealing with complex scenes, and the applicability of the algorithm is limited. In addition, the computational complexity of traditional multi-scale image fusion algorithms is also high, which leads to slow operation of the algorithms and is not suitable for real-time or large-scale applications. In addition, the algorithms or metrics on which these methods are based are specialized for specific fusion tasks, and their generalization ability is not outstanding.

### 2.2. Deep Learning-Based Image Fusion Algorithms

Deep learning is an important research area in the field of image processing, and significant progress has been made in image fusion techniques in recent years, which has now become a research hotspot in the field. Liu Y et al. proposed a CNN-based image fusion method which utilizes convolutional networks to obtain the feature weights, and according to the similarity of the local information of the source image, the fusion rules of the corresponding weighting and maximization are selected [25]. H. Li et al. proposed [26] to fuse infrared and visible images by using a pre-trained VGG network, which realizes the extraction and fusion of multilayer depth features of the source image. Prabhakar et al. [27] proposed an image fusion depth algorithm, DeepFuse, and realized the unsupervised training of the network by a reference-free image evaluation metric. H. Li et al. [28] used ResNet-50 to extract the depth features of the source image and fused them. The ResNet-50 network feature extraction capability is more powerful than that of the VGG network, with better fusion results. The disadvantage is that the model cannot adaptively select or fuse depth features. Later, they proposed a new fusion algorithm, DenseFuse [29]. DenseFuse is a self-coding network consisting of an encoder and a decoder. The network encoder is used to extract features, and the decoder is used to generate the fused image. DenseFuse achieves better fusion results, but the fusion strategy still needs to be manually designed, which adds a certain amount of computational complexity. Ma et al. [30] proposed FusionGAN, an algorithm for the fusion of infrared and visible images using a generative adversarial approach. The algorithm consists of a generator and an adversary, where the generator is used to generate the fused image and the purpose of the discriminator is to force the fused image to retain more details and texture information from the visible image. This provides new ideas for image fusion tasks. Liu et al. [31] proposed a generalized network fusion framework, IFCNN. IFCNN uses clear images and blurred maps as labeled data to train the network and then uses different feature fusion strategies for different tasks to achieve multi-task image fusion. Li et al. designed a novel infrared and visible fusion network, NestFuse [32], which embeds spatial and channel attention mechanisms, extracts multi-scale features of the source image through a decoder, and uses a fusion strategy to fuse the scale features of different dimensions and then reconstructs them using the decoder in order to obtain the fused image, with a significant improvement in fusion performance.

In general, deep learning-based image fusion methods improve the limitations of traditional algorithms to a certain extent, but they also face some challenges:How to retain the detailed texture information and global features in the source image in a more complete way, to solve the problem of balancing the detailed and global information in the image fusion process and to improve the overall quality of the fused image.Although the deep learning model can learn the feature representation of the image through end-to-end training, there is still the problem that some information is lost in the image fusion task, which ignores the local representation information of the source image.

## 3. Proposed Method

In this section, we present the designed fusion network for infrared and visible images in detail. The network structure, loss function, and fusion strategy are presented.

### 3.1. Network Architecture

The overall architecture of the fusion network (DCFNet) proposed is shown in Figure 1. It is an encoder–decoder-based network structure, which consists of three main parts: encoder, fusion layer, and decoder.

As shown in Figure 1, the encoder consists of a DWT module, a convolutional layer, three bottleneck residual blocks, and a coordinate attention (CA) module. The encoding network is designed with a low-frequency global branch and a high-frequency detail branch to handle low-frequency features and high-frequency features, respectively. The parameters of the convolutional modules within different branches are kept consistent except for the different input and output channels. The source image is directly input into the encoder and processed by the DWT module to obtain shallow low-frequency features and high-frequency features. The size of the convolution kernel of the convolutional layer is 3 × 3, aiming at extracting rough frequency features. The structure of the residual block is shown in Figure 2; each residual module consists of three convolutional layers constituting the residual path and the direct mapping path, which is designed to improve the training convergence of the network and extract more significant intermediate frequency features. In the residual block, all the convolutional layers use 1 × 1 convolutional kernels, except the middle convolutional layer, which uses 3 × 3 convolutional kernels; in addition, all the convolutional layers use LeakyReLU as the activation function. The structure of the coordinate attention module is shown in Figure 3. The coordinate attention module [33] enhances the network’s ability to extract key features in infrared and visible images by focusing on inter-channel relationships and spatial location information, thus highlighting important features and suppressing useless features.

Unlike the encoder, the decoder comprises an IDWT module followed by three convolutional layers. These layers are applied sequentially, with the number of output channels gradually being reduced to one, ultimately producing the fused image. The first two convolutional layers utilize the LeakyReLU activation function, while the final layer employs the Sigmoid activation function to map output values to the range of 0 to 1. To prevent information loss during the fusion process, all convolutional layers in the network use the same padding, ensuring that the fused image retains the same dimensions as the source image.

### 3.2. DWT Image Decomposition and IDWT Image Reconstruction

DWT is a signal analysis technique that is widely used in the field of signal processing and image compression [34]. DWT can decompose an image into different structures, including low-frequency and high-frequency images. High-frequency images are directionally selective and are classified as horizontal, vertical, and oblique, while low-frequency images are strongly correlated and contain most of the information of the image. This ensures that more structural information is retained in image processing [35]. Figure 4 shows the 2D discrete wavelet transform decomposition process of the image, where the right part shows the sign of each sub-band of the wavelet coefficients. In the figure, LL, LH, HL, and HH are four equal-size sub-band images representing low-frequency information and high-frequency information in the horizontal, diagonal, and vertical directions, respectively.

Inspired by the applicability of the generic DWT/IDWT network layer for image denoising [36], the discrete wavelet transform for image decomposition, DWT, preserves the object structure and data details, while IDWT accurately recovers the original data. The image is converted into the frequency domain through the DWT layer to achieve the adaptive learning of different frequency components, followed by the IDWT layer to complete frequency feature fusion and reconstruction. The reversibility of the discrete wavelet transform ensures that the network fully retains the detailed texture information of the image during the feature extraction process and avoids the problem of information loss during the calculation of the network model. Due to the compatibility of the DWT and IDWT layers with the convolutional layer, flexible learning in the frequency space is allowed, which improves the adaptability and efficiency of the model. Common wavelet functions include Daubechies, Morlet, and Haar. we chose the Haar wavelet and set the number of decomposition layers to one. Figure 5 shows the feature visualization of the input image decomposed by the DWT layer.

The DWT decomposition steps and IDWT reconstruction steps are as follows:(1)$$I_{L}\left(m,n\right)=\frac{1}{\sqrt{MN}}\sum_{x=0}^{M-1}\sum_{y=0}^{N-1}I\left(x,y\right)\varphi_{m,n}\left(x,y\right)$$
(2)$$I_{H}\left(m,n\right)=\frac{1}{\sqrt{MN}}\sum_{x=0}^{M-1}\sum_{y=0}^{N-1}I\left(x,y\right)\psi_{m,n}^{i}\left(x,y\right)$$

Above, *I* denotes the source image, *M* and *N* denote the length and width of the source image, $I_{L}\left(m,n\right)$ denotes the low-frequency feature map, and $I_{H}\left(m,n\right)$ denotes the high-frequency feature map. $\left(x,y\right)$ denotes the null-domain coordinate value of the pixel point in the source image; $\left(m,n\right)$ denotes the null-domain coordinate value of the pixel point in the frequency image; $\varphi_{m,n}\left(x,y\right)$ denotes the scale function, whose expression is shown in (Equation 3); $\psi_{m,n}^{i}\left(x,y\right)$ denotes the wavelet function, whose expression is shown in (Equation 4)
(3)$$\varphi_{m,n}\left(x,y\right)=2^{j/2}\varphi \left(2^{j}x-m,2^{j}y-n\right)$$
(4)$$\begin{array}{cc} \psi_{m,n}^{i}\left(x,y\right) & =2^{j/2}\psi^{i}\left(2^{j}x-m,2^{j}y-n\right)\\ i & =H,V,D \end{array}$$

The formula for the IDWT inverse transform reconstruction of the source image is defined as
(5)$$F\left(x,y\right)=\frac{1}{\sqrt{MN}}\sum_{m}\sum_{n}f_{L}\left(m,n\right)\varphi_{m,n}\left(x,y\right)+\frac{1}{\sqrt{MN}}\sum_{m}\sum_{n}f_{i}\left(m,n\right)\psi_{m,n}^{i}\left(x,y\right)$$
where $f_{L}\left(m,n\right)$ represents the low-frequency frequency image, $f_{i}\left(m,n\right)\;\;\;i=\left\{H,V,D\right\}$ represents the corresponding high-frequency frequency image, and F(x,y) represents the fused image.

### 3.3. Fusion Strategy

After completing the training phase in Figure 1a with the encoder and decoder networks fixed, we start the task of fusing IR and visible images. As shown in Figure 1b, in the testing phase, a pair of infrared and visible images are first input into the encoder; then, the extracted feature mapping is input into the fusion layer for fusion, and the $l_{1}-norm$ fusion strategy is selected in this phase; finally, the fused features are input into the decoder to reconstruct the fused images. The fusion process equation is defined as
(6)$$\begin{array}{cc} \Phi_{F}^{L} & =F(\Phi_{I}^{L},\Phi_{V}^{L})\\ \Phi_{F}^{H} & =F(\Phi_{I}^{H},\Phi_{V}^{H}) \end{array}$$
where $\Phi_{V}^{L}\;$ and $\Phi_{V\;}^{H}$ represent visible low-frequency features and visible high-frequency features, respectively. $\Phi_{I}^{L}$ and $\Phi_{I}^{H}$ represent infrared low-frequency features and infrared high-frequency features, respectively. $\Phi_{F}^{L}$ and $\Phi_{F}^{H}\;$ represent low-frequency fusion features and high-frequency fusion features, respectively. F(·) stands for the $l_{1}-norm$ fusion strategy. The $l_{1}-norm$ fusion strategy is shown in Figure 6.

For both the fusion of low-frequency features and the fusion of high-frequency features, the $l_{1}-norm$ fusion strategy is adopted. Firstly, the initial activity level map $C_{k}$ is calculated. m∈1,2,3,…,M,M=64 represents the number of feature maps, and $\phi_{k}^{m}$ represents the features extracted by the encoder, while *k* indicates that the inputs are an infrared image and a visible-light image, where k∈{1,2}, as shown in (Equation 7).
(7)$$C_{k}\left(x,y\right)={∥\phi_{k}^{1:M}\left(x,y\right)∥}_{1}$$

The final activity level map is then calculated by using block-based averaging operations as shown in (Equation 8).
(8)$${\hat{C}}_{k}\left(x,y\right)=\frac{\sum_{a=-r}^{r}\sum_{b=-r}^{r}C_{k}\left(x+a,y+b\right)}{{\left(2r+1\right)}^{2}}$$

The final fused feature map $f^{m}$ can be obtained by Equations (Equation 9) and (Equation 10).
(9)$$w_{k}\left(x,y\right)=\frac{{\hat{C}}_{k}\left(x,y\right)}{\sum_{n=1}^{k}C_{n}\left(x,y\right)}$$
(10)$$f^{m}\left(x,y\right)=w_{1}\left(x,y\right)\times \phi_{1}^{m}\left(x,y\right)+w_{2}\left(x,y\right)\times \phi_{2}^{m}\left(x,y\right)$$

### 3.4. Loss Function

In order to ensure that the training process achieves the desired effect, a reasonable loss function needs to be designed. In the image fusion process, the loss function mainly evaluates the difference between the output image and the input image and, through parameter optimization, guides the output image of the network model to match the input image structurally, retaining as much detail information as possible in the image, to continuously improve the performance of the network model as well as to generate high-quality fused images. This chapter uses a weighted loss consisting of gradient loss $L_{Grd}$, pixel loss $L_{P}$, and structural similarity loss function $L_{SSIM}$; λ and β are the weighting coefficients, which together constrain the gradient, structural, and pixel errors in the reconstruction process. Its total loss expression is defined as
(11)$$L_{\mathrm{total}}=L_{P}+\lambda L_{\mathrm{SSIM}}+\beta L_{\mathrm{Grd}}$$

(1) Pixel loss $L_{P}$

$L_{P}$ is used to calculate the pixel error between the output image and the input image. Its calculation is shown in (Equation 12).
(12)$$L_{P}=\mathrm{MSE}\left(O,I\right)=\frac{1}{N}\sum_{n=1}^{N}{\left(O_{n}-I_{n}\right)}^{2}$$
where *I* denotes the input image, *O* denotes the output image, *N* denotes the number of samples, and MSE(O,I) is the mean square error function between *O* and *I*.

(2) Loss of structural similarity $L_{SSIM}$

$L_{SSIM}$ is a metric for evaluating the structural similarity between the input image and the output image, with a value ranging from 0 to 1. The specific function expression is shown below.
(13)$$L_{\mathrm{SSIM}}=1-\mathrm{SSIM}\left(O,I\right)$$
(14)$$\mathrm{SSIM}\left(O,I\right)=\frac{\left(2\mu_{O}\mu_{I}+c_{1}\right)\left(2\sigma_{OI}+c_{2}\right)}{\left(\mu_{O}^{2}+\mu_{I}^{2}+c_{1}\right)\left(\sigma_{O}^{2}+\sigma_{I}^{2}+c_{2}\right)}$$
where $\mu_{O}$ and $\mu_{I}$ denote the mean value of the image; $\sigma_{O}\;$ and $\sigma_{I}\;$ denote the variance values of the output image and the input image, respectively; $\sigma_{OI}$ denotes the covariance between the input image and the output image; and $c_{1}4$ and $c_{2}$ are constants used for the calculation of the stable denominator.

(3) Gradient losses $L_{Grd}$

$L_{Grd}$ uses the gradient information of the model parameters to measure the performance of the model by calculating and comparing the gradient difference of the local structural information between the output image and the input image. The loss function expression is shown in (Equation 15).
(15)$$L_{\mathrm{Gra}}=\mathrm{MSE}\left(\mathrm{Gradient}\left(O\right),\mathrm{Gradient}\left(I\right)\right)$$
where Gradient(·) denotes the operation of enhancing the edge and detail information in the image by using the Laplace transform. The Laplace operation expression is shown in (Equation 16).
(16)$$\begin{array}{c} \nabla^{2}f\left(x,y\right)=\frac{\partial^{2}f\left(x,y\right)}{\partial x^{2}}+\frac{\partial^{2}f\left(x,y\right)}{\partial y^{2}}\\ \approx [f(x+1,y)+f(x-1,y)+f(x,y+1)+f(x,y-1)]-4f(x,y) \end{array}$$
where $f\left(x,y\right)$ denotes the gray value of the image and $\nabla^{2}$ denotes the Laplace operator.

## 4. Experiments

In this section, the experimental setup, dataset, and evaluation metrics are introduced. Experiments are then conducted on public datasets to compare this paper’s method with seven representative fusion methods, including DWT [37], FusionGAN [30], GTF [38], CNN [25], DenseFuse [29], DIDFuse [39], and NestFuse [32]. The SwinFuse [40] and MUFusion [41] comparison algorithms are based on publicly available code implementations with parameters set according to the experiments in the original paper. In addition, generalization experiments are conducted to evaluate the generalization performance of the algorithm in this paper.

### 4.1. Experimental Setup

#### 4.1.1. Datasets and Pre-Processing

We chose to conduct our experiments on three commonly used public datasets, TNO [42], NIR [43], and RoadScene [32]. The TNO dataset, as a classical image fusion public dataset, shows multispectral nighttime images captured by different camera systems. The dataset contains 1000 infrared and visible images covering different military scenarios of soldiers, buildings, vehicles, etc. The NIR dataset was captured by using individual exposures with a modified digital SLR camera. The dataset consists of 477 infrared and visible image pairs, including nine types of natural scenes, such as countryside, street, mountain, and forest. The RoadScene dataset is derived from FLIR videos which were accurately clipped, aligned, and pre-processed with background thermal noise to solve the problems of fewer pairs of images in the baseline dataset, low spatial resolution, and lack of detailed information in the infrared images and to ensure the quality of the images. The dataset contains 221 pairs of aligned visible and infrared image pairs, covering a rich variety of scenes, including representative scenes of roads, vehicles, and pedestrians.

The Adam optimizer is used in the training phase to train the model, with the batch size batch set to 12, the learning rate parameter set to 1 × 10^−4^, and the parameters λ and β in the loss function empirically set to 100 and 10, respectively. The number of training rounds is 300. All fusion methods are implemented on the Windows 10 operating system and trained on a computer equipped with an AMD Ryzen 7 5800H CPU and NVIDIA RTX3060 GPU for training. The network model utilizes the Pytorch framework. The DCF training steps (described in the accompanying pseudo-code) encapsulate the modeling algorithm as described below (Algorithm 1).
**Algorithm 1** Training steps for DCF 1:**Input:** image *T* in the training set 2:**Output:** reconstructed image *R* 3:Initialize network parameters, Adam optimizer parameters, number of iterations *M*, learning rate, and loss function related parameters 4:Set the batch size *N* and input image size for training 5:**for** *M* epochs **do** 6:    Data loading, computation of the forward network output image based on the network model 7:    Select *b* input images $\left\{T^{1},T^{2},\cdots ,T^{b}\right\}$ into the encoder, and utilize the DWT layer image decomposition to generate low-frequency global features $\left\{S_{l}^{1},S_{l}^{2},\cdots ,S_{l}^{b}\right\}$ as well as high-frequency detailed features $\left\{S_{h}^{1},S_{h}^{2},\cdots ,S_{h}^{b}\right\}$ 8:    The frequency features are inverted by the IDWT layer of the encoder, and the final encoder outputs the reconstructed image $\left\{R^{1},R^{2},\cdots ,R^{b}\right\}$ 9:    The forward output reconstructed image is substituted with the input image into the loss function $L_{\mathrm{total}}$, which is back-propagated and updated by Adam with the network parameters *d*10:    The formatted output includes the current epoch, current step, loss value, and current learning rate11:**end for**12:Save trained weights and results13:**return** Output

#### 4.1.2. Evaluation Indicators

Subjective evaluation is a qualitative analysis that perceives how good or bad the image fusion result is by direct observation of the image by the observer’s eye visual system. When the differences between the fused images are small, it may not be possible to obtain accurate evaluation results by relying only on subjective evaluation, so combining objective evaluation methods can improve the accuracy of the evaluation results. Objective evaluation methods do not rely on individual subjective feelings but are based on specific attributes of the fused images and pre-determined criteria. Therefore, researchers can conduct a comprehensive assessment of fused images by combining different evaluation indexes. Five mainstream evaluation metrics were selected for fused image quality evaluation, including MI, SF, EN, SSIM, and PSNR. They are defined as follows:(17)$$MI=MI^{AF}+MI^{BF}$$
where *A* and *B* represent a pair of source images and *F* represents the fused image. ${MI}^{AF}$ and ${MI}^{BF}$ denote the amount of mutual information between the source images *A* and *B* and the fused image *F* with the amount of mutual information between them.
(18)$$MI^{AF}=\sum_{f=0}^{L}\sum_{a=0}^{L}p^{AF}\left(a,f\right)\log_{2}\left(\frac{p^{AF}\left(a,f\right)}{p^{A}\left(a\right)p^{F}\left(f\right)}\right)$$
(19)$$MI^{BF}=\sum_{f=0}^{L}\sum_{a=0}^{L}p^{BF}\left(b,f\right)\log_{2}\left(\frac{p^{BF}\left(b,f\right)}{p^{B}\left(b\right)p^{F}\left(f\right)}\right)$$
where $p^{A}\left(a\right)$, $p^{B}\left(b\right)$, and $p^{F}\left(f\right)$ denote the edge probability density functions of the three images, and $p^{AF}\left(a,f\right)$ and $p^{BF}\left(b,f\right)$ denote the joint probability density functions of the source images *A* and *B* and the fused image *F*.
(20)$$SF=\sqrt{RF^{2}+CF^{2}}$$
(21)$$RF=\sqrt{\frac{1}{MN}\sum_{i=1}^{M}\sum_{j=2}^{N}{\left[F(i,j)-F(i,j-1)\right]}^{2}}$$
(22)$$CF=\sqrt{\frac{1}{MN}\sum_{j=1}^{M}\sum_{i=2}^{N}{\left[F(i,j)-F(i-1,j)\right]}^{2}}$$
where *M* and *N* denote the width and height of the fused image. RF and CF denote the spatial row frequency and spatial column frequency, respectively, of the fused image; $F\left(\cdot \right)$ denotes the gray value of the corresponding pixel point of the fused image.
(23)$$EN=-\sum_{l=0}^{l-1}p_{l}\log_{2}p_{l}$$
where *l* is the gray level of the image and $p_{l}$ is a normalized histogram of the image gray level indicating the probability of a pixel point value to appear in the image.
(24)$$SSIM\left(x,y\right)=\frac{\left(2\mu_{x}\mu_{y}+C_{1}\right)\left(2\sigma_{xy}+C_{2}\right)}{\left(\mu_{x}^{2}+\mu_{y}^{2}+C_{1}\right)\left(\sigma_{x}^{2}+\sigma_{y}^{2}+C_{2}\right)}$$
where SSIM takes values ranging from 0 to 1, the closer the value is to 1, the higher the similarity between the fused image and the source image, and the better the fusion effect.
(25)$$MSE=\frac{1}{H\times W}\sum_{i=1}^{H}\sum_{j=1}^{W}{\left(I(i,j)-O(i,j)\right)}^{2}$$
(26)$$PSNR=10\times \log_{10}\left(\frac{MAX^{2}}{MSE}\right)$$
where *H* and *W* are the height and width of the image, respectively, $I\left(i,j\right)\;$ and $O\left(i,j\right)\;$ represent the pixel values of the corresponding points of the image, respectively, and MSE denotes the mean square error between image *I* and image *O*. MAX denotes the maximum pixel value of the image.

### 4.2. Comparison Test of TNO Datasets

In order to fully evaluate the performance of this paper’s algorithm (DCF), this method is compared with nine other methods on the TNO dataset.

#### 4.2.1. Subjective Qualitative Comparisons

In order to clearly compare the experimental results, the key areas in the three sets of scenes—characters, soldiers, and dark fortresses—were specially selected, and these areas were marked with red boxes to be enlarged for the purpose of demonstrating the differences in fusion effects in detail. As shown in Figure 7, GTF mainly retains the information of the infrared image but has defects in maintaining the detailed texture information of the visible image, resulting in the sky color appearing darker. CNN also suffers from insufficient texture extraction of the visible image, as well as the loss of the character target in the infrared image. DWT retains the detailed texture information of the visible image better but is accompanied by artifacts. DenseFuse is more affected by noise, resulting in a lower overall contrast of the fused image and insufficient extraction of detailed information about people, trees, and grass. The fusion results of FusionGAN and DIDFuse are poor in visual effect, with blurred edges, large distortion, and artifacts in the image. The key features of MUFusion are more completely retained, but there is obvious noise information. NestFuse, SwinFuse, and DCF have the best overall visual effect, among which the fusion image of DCF is more outstanding in terms of contrast and clarity of character and target information. The DCF method performs well in retaining the detailed texture information of the visible and infrared images, there is no obvious noise nor artifact in the fusion results, and the overall quality of the fused image is better.

As shown in Figure 8, FusionGAN, DIDFuse, and NestFuse have blurred contours of the soldiers concealed by smoke and lack the target information in the infrared image, with FusionGAN losing a lot of texture information and presenting edge blurring and noise contamination. DenseFuse and CNN, although they retain the target information in the infrared image and the contours of the soldiers in the smoke are relatively clear, perform insufficiently in extracting texture detail information in visible-light images. SwinFuse’s fusion quality is relatively good, but this method introduces a small amount of noise information. With MUFusion, soldier information is retained more completely, but the contrast is poor. The fusion result of this paper’s algorithm, DCF, shows that it not only clearly extracts the soldiers’ silhouette information but also accurately retains the details of smoke, and the overall image quality is significantly better than in other methods. As shown in Figure 9, this paper’s algorithm, DCF, not only effectively highlights the dark fortress target information in the scene but also retains the detailed texture information of the bushes more adequately. In contrast, GTF, FusionGAN, and CNN show obvious blurring and loss of detail information when dealing with background regions such as bushes. Although NestFuse, DenseFuse, and DWT retain the detail information of bushes in visible images better, the target information extraction for infrared images is not sufficient, and it is difficult to distinguish dark forts and bushes, which is not conducive to military reconnaissance and combat missions. The colors extracted from the bushes by DIDFuse are not of the same shade, and the overall aesthetic of the image is not good. SwinFuse’s and MUFusion’s fusion visual effects are good, and the key information extraction is more adequate, but there is a loss of texture details.

In summary, DCF is significantly better than the other nine algorithms in subjective qualitative comparisons, and DCF is able to effectively extract texture information from visible images and target information from infrared images, with the best overall effect of fused images.

#### 4.2.2. Objective Quantitative Comparisons

Due to the limitation of human eye perception for subjective evaluation, in order to improve the accuracy of the experimental results, this subsection chooses to conduct objective quantitative comparison of the fusion results of the eight algorithms, and a comparison of the objective evaluation metrics of the fusion results is shown in Table 1. Figure 10 illustrates the line graphs of the scores of the ten fusion results on five indicators, and the legends correspond to different fusion methods. The image fusion algorithms compared are based on publicly available code, and all data in the table are from measurements of the fused images.

As can be observed in Table 1, the fusion algorithm DCF performs optimally in the four metrics of PSNR, SSIM, SF, and MI, which indicates that DCF achieves significant advantages in fused image distortion, information richness, correlation with the source image, and clear texture details. Despite the slightly poor performance in the MI metric, it still ranks second. The results of the objective indicators show that compared with other fusion algorithms, DCF fully retains the detailed information of the source image, with clear contours and no obvious distortion nor artifacts, and the fused image shows better visual quality and higher information fidelity, which indicates that algorithm fusion is more effective.

Comprehensive subjective and objective analyses show that this paper’s algorithm, DCF, demonstrates excellent fusion performance on the TNO dataset, with obvious advantages over several other image fusion algorithms, as it not only maintains a high degree of image quality and information integrity but also exhibits an efficient ability to preserve key visual features and details.

### 4.3. Comparison Test of NIR Datasets

In order to verify the effectiveness of the algorithms in this chapter, this paper conducts comparative experiments on the NIR dataset to evaluate the fusion performance of different fusion algorithms when dealing with natural landscapes.

#### 4.3.1. Subjective Qualitative Comparisons

Figure 11 shows the fusion results of DCF and the other nine fusion algorithms on the NIR dataset. From top to bottom, for infrared images and visible images, the fusion results of DWT, DenseFuse, GTF, CNN, NestFuse, FusionGAN, DIDFuse, SwinFuse, MUFusion, and DCF are shown. A comparison of the results of the fused images shows that GTF and FusionGAN have obvious color distortion in all the five sets of images from Figure 11a to Figure 11d, especially in the target information of tree trunks next to the street and clouds in the sky. Among them, the fusion result of FusionGAN also has obvious artifacts, and the overall fusion effect is poor. The fused images of Dense Fuse have blurred edge contours, and the target information of the trees in the group of images in Figure 11a shows artifacts, which affects the visual effect. The overall contrast of CNN is low, the clarity is poor, and the details of the images are not prominent enough. The overall contrast of DIDFuse is not high, and the detail retention is also low. Nest Fuse has high image clarity, but the ability to retain detailed texture information in visible-light images is poor. SwinFuse’s image quality is relatively good, but not enough texture detail information is retained. MUFusion’s image clarity is not high, and the outline is blurred. Compared with the other algorithms, the fused image of DCF has moderate clarity and brightness, which is more in line with the visual habit of the human eye, and is able to better retain the texture details of the visible image and the target information of the infrared image; further, the overall quality of the fusion image is better, which demonstrates higher image quality and information integrity.

#### 4.3.2. Objective Quantitative Comparisons

Since there is little subjective visual variability among the fused images, it is not possible to accurately judge the merits of the fused images based on visual observation alone. Therefore, five objective evaluation indexes, namely, peak signal-to-noise ratio (PSNR), structural similarity index (SSIM), spatial frequency (SF), mutual information (MI), and entropy value (EN), were selected to objectively evaluate the fusion results of different algorithms. The objective evaluation metrics of different fusion algorithms on the NIR dataset are shown in Table 2. Among them, the optimal values are shown in red bold font, and the sub-optimal values are shown in black bold font.

As can be seen from Table 2, the fusion results of the algorithms in this chapter are optimal in the PSNR, SF, and EN metrics and also rank second in the SSIM and MI metrics with a small disadvantage. This indicates that the algorithm retains the source image information more fully, expresses the feature information more accurately, and has better contrast and brightness and that the visual effect of the fused image is better. Combining the results of subjective evaluation and objective indicators, the algorithm has good fusion performance and high quality of fused images, which verifies its superiority.

### 4.4. Generalization Experiments

To further validate the generalization performance of the algorithms, we conduct generalization experiments on the RoadScene dataset to evaluate the fusion performance of different fusion algorithms in rich road scenarios.

#### 4.4.1. Subjective Qualitative Comparisons

Figure 12 shows the fusion results of nine different algorithms on the RoadScene dataset. From top to bottom, the fusion results of infrared images and visible-light images of DWT, DenseFuse, GTF, CNN, NestFuse, FusionGAN, DIDFuse, SwinFuse, MUFusion, and DCF are plotted. In order to clearly compare the experimental results, the regions with large differences in fusion results are marked with red boxes. From the comparison of the result graphs of the fused images, it can be seen that the DenseFuse, DWT, CNN, and GTF images have low clarity, missing texture details, insufficient target information extraction, artifacts, and blurring of contour edges. The FusionGAN images have low clarity, significant missing image detail information, blurring of the edges, and distortions. The DenseFuse infrared images show better target information extraction, but edge information and detailed texture information are severely missing. DIDFuse more complete retains the source image information, but there is the problem of uneven brightness, and the fusion image contrast is low. The NestFuse, SwinFuse, and MUFusion images are visually better, the fusion effect is relatively good, and infrared target information retention is clearer, but the image detail information extraction ability is weaker. In contrast, the target information contour edges in the DCF infrared image are clearer, while the texture detail information of the visible image is more fully retained, and the visual effect is excellent. For example, in Figure 12a, the road pedestrian target information is clear, and the background road information is more fully extracted. In Figure 12b,d, the target information of vehicles on the road is the clearest; in Figure 12c, detailed texture information, such as the “FLIR” font and the “6769” number, is the clearest, and the image contrast is high, so the fusion quality is relatively better.

#### 4.4.2. Objective Quantitative Comparisons

Due to the limitations of subjective evaluation of the human eye, in order to improve the accuracy of the experimental results, a total of five objective evaluation metrics, PSNR, SSIM, SF, MI, and EN, were selected for objective quantitative evaluation in this paper. The comparison results of the objective metrics on the RoadScene dataset are shown in Table 3, where all the data metrics are the results of the measurement and evaluation of the fused images. The optimal values are shown in red bold font, and the sub-optimal values are shown in black bold font.

As can be seen from Table 3, the algorithm proposed in this chapter performs optimally in all three metrics, PSNR, SF, and EN. Although, in the SSIM and MI metrics, it does not achieve the optimal value, in these metrics, it ranks second by a slight difference. The objective evaluation results show that the algorithm in this chapter achieves a significant improvement in the quality and performance of the fused images compared with the other nine algorithms. Combining the results of subjective and objective metrics evaluation, the algorithm in this chapter has excellent fusion performance on the RoadScene dataset.

### 4.5. Ablation Experiments

#### 4.5.1. Comparison of Different Fusion Strategies

Ablation experiments are conducted based on the additive and $l_{1}-norm$ feature fusion strategies, and the results are compared. The fusion results of different fusion strategies are shown in Figure 13. No changes are made to the training environment for the purpose of environment configuration, dataset, and hyperparameter settings.

As shown in Figure 13, both fusion strategies are able to accomplish the task of fusing infrared and visible images relatively well. However, in terms of the performance of the target information in the two pairs of images, the $l_{1}-norm$ fusion strategy retains the target information with higher brightness, the structure of the house as well as the outline of the smoke is clearer, and the contrast is higher. Compared with the fusion results of the $l_{1}-norm$ fusion strategy, the fused images of the additive strategy appear blurred in edges and details, with poor subjective visualization.

In order to improve the accuracy of evaluating the effect of fusion strategies, this section selects a total of five objective evaluation indexes, namely, PSNR, SSIM, SF, MI, and EN, to objectively and quantitatively evaluate the fusion results of different fusion strategies, and the objective evaluation results of different fusion strategies are shown in Table 4.

From the data in Table 4, it can be seen that the $l_{1}-norm$ fusion strategy achieves optimal performance in the four indexes of SSIM, SF, MI, and EN, and although it is slightly lower than the additive fusion strategy in the PSNR index, the visual gap is not significant. Overall, the fused images obtained by the $l_{1}-norm$ fusion strategy are superior in information retention and contrast, and the fusion effect is excellent. The combined results of subjective qualitative evaluation and objective quantitative evaluation indicate that the $l_{1}-norm$-based fusion strategy is superior to the addition-based fusion method in terms of performance. Therefore, the $l_{1}-norm$ fusion strategy is selected as the feature fusion method for the algorithm in this chapter.

#### 4.5.2. Comparison of Different Ablation Network Models

In order to verify the effect of each module on the network structure, this study constructs three ablation models by performing ablation experiments on different structural variants of the network and compares the resulting experimental results. In order to ensure the fairness of the ablation model experiments, no changes are made to the training environment, such as environment configuration, dataset, and hyperparameter settings, during the experiments, except for the different choices of ablation modules. The experimental results are shown in Figure 14.

Detailed descriptions of the three ablation models: (c) DCF-ND: Compared with DCF, the DWT and IDWT modules are missing in the network structure; there is no branching network designed for high-frequency feature and low-frequency feature branching; and a 3 × 3 convolutional layer is directly used for feature extraction. (d) DCF-NC: Compared with DCF, the CA coordinate attention module is missing in the network structure. (e) DCF: the complete fusion network, where no adjustment is made to the network structure.

From the fusion results shown in Figure 14c–e, it can be seen that DCF-ND can retain the salient information in the IR image; however, this is not sufficient for the extraction of the texture detail information in the visible image, and the overall contrast is low. DCF-NC lacks the coordinate attention mechanism, and there is a lack of retention of key information. In contrast, DCF performs better in retaining IR target information and visible-light detail texture information in the fused image. For example, in the first row of the image experiments, in Figure 14c, the clouds in the fused image are missing, and in Figure 14e, the clouds in the fused image contain more detailed information and have higher clarity than the clouds in the fused image in Figure 14d. In the second and third rows of the image experiments, the fused image in Figure 14e has the most obvious infrared target information, along with rich detail information. Therefore, the subjective analysis shows that the fused image generated by DCF is of better quality.

In order to improve the accuracy of the assessment of the fusion effect of different ablation models, a total of five objective evaluation indexes, namely, PSNR, SSIM, SF, MI, and EN, were selected to evaluate the fusion results of different ablation models. The objective evaluation results corresponding to the fusion results of each ablation model are shown in Table 5.

As can be seen from Table 5, DCF-ND has the lowest score in the SF metric and the second lowest score in EN, which indicates that the fused image of this model has the lowest detail retention and that the high-frequency detail information is not sufficiently extracted. DCF-NC has the lowest scores in MI and EN, which indicates that the fusion result of this model has less information, as well as missing inter-channel information and position information, and that the fused image information correlation is weaker, with poor fusion effect. In contrast, DCF has optimal values in the five indexes of PSNR, SF, MI, EN, and SSIM, indicating that DCF can retain the key feature information of infrared images and visible images, the source image information retention is high, the fusion result performance is better, and fusion images of high quality can be obtained. Following these comprehensive subjective and objective analyses, the ablation experimental results show that the fusion algorithm with the addition of the DWT and IDWT modules and the coordinate attention mechanism has better performance, which verifies the effectiveness of the algorithm in this paper.

### 4.6. Addition of Noise

In order to explore the performance capability of our algorithm under difficult conditions, we add a set of comparison experiments in this subsection. Considering the relatively good quality of the existing test dataset, in the experiments, we artificially added Gaussian noise and pepper noise, which are difficult to be distinguished by the human eye, to the visible images and fused them; then, we selected several methods for comparison, as shown in Figure 15. The fusion results of the three kinds of fusion methods, namely, NestFuse, SwinFuse, and MUFusion, are shown in Figure 15. In terms of noise reduction effect, through comparison, we found that the noise in the fusion result of DCF is the smallest, indicating higher performance. Thanks to the excellent performance of DWT in image denoising, the saliency target of the fused image is very obvious, and more comprehensive scene information is obtained, which not only includes highlighting the target and the background details but also includes highlighting the details of the target’s texture. The experiments show that the denoising effect of the algorithm proposed in this paper is significant and can be applied to complex conditions. This subsection is only a simple attempt, and we will further improve our research in the future based on images acquired under more extreme weather conditions.

### 4.7. Efficiency Comparison

Running efficiency is also an important factor in evaluating the performance of a model. In this subsection, we continue to test the computational efficiency of our DCF, and we provide the average runtime of 10 different methods on the TNO, NIR, and RoadScene datasets, as shown in Table 6. In general, we can see that deep learning-based algorithms have a large advantage in computational efficiency at runtime because they are accelerated by using GPUs. In contrast, traditional methods take longer to produce fused images. Specifically, DWT needs to decompose the source image into a low-frequency part and a high-frequency part by wavelet transform representation, so it takes more time to achieve fusion. In terms of results, our computational efficiency is second only to DenseFuse, but our DCF is competitively computationally efficient, and the network makes the DWT and IDWT layers compatible with the convolutional layers, allowing for flexible learning in the frequency space, which improves the model’s adaptability and efficiency. Therefore, we can conclude that our DCF has better fusion performance, stronger generalization ability, and competitive computational efficiency, further proving the effectiveness of the designed network.

## 5. Conclusions

In this paper, a fusion algorithm for infrared and visible-light images based on discrete wavelet transform and convolutional neural network is proposed. The DWT frequency extraction module is embedded in the coding network to extract the high-frequency and low-frequency features of the infrared and visible images; then, the high-frequency detailed feature branching network and the low-frequency global feature branching network process the extracted high-frequency features and low-frequency features, respectively, in order to realize the comprehensive extraction of the features at each frequency. In addition, a residual module is designed in the coding network to enhance the feature extraction capability of the network. Meanwhile, the coordinate attention mechanism is introduced into the coding network to enhance the ability of capturing global information and feature characterization. The feature fusion process uses the $l_{1}-norm$ fusion strategy to fuse the features extracted by the encoder. During the training process, a weighted loss function consisting of pixel loss, gradient loss, and structural loss is used to constrain the network, which in turn improves the quality of the fused images. The fused features are reconstructed by the IDWT frequency feature reconstruction module; then, the fused image is output through the decoder. The experimental results show that the algorithm proposed in this paper displays excellent performance in both subjective and objective evaluations and performs optimally on the RoadScene dataset in the three metrics of PSNR, SF, and EN, which are 4 percent, 1.8 percent, and 1.78 percent better than the second place, respectively. The superiority of the algorithm is effectively verified. It also has good generalization ability, which can be applied to other datasets, and good denoising performance, which can be applied to tasks under complex conditions.

Additionally, it is important to recognize that the wavelet transform is inherently computationally intensive, and integrating it with a CNN can significantly increase the overall computational complexity and runtime, particularly when processing high-resolution images. In addition, the wavelet transform requires learning additional parameters, which may increase the total number of parameters in the model, which may also lead to increased memory and computational requirements during training and inference. In future work, we will adapt and optimize the algorithm to enhance the compatibility of wavelet transform and CNN, design a general fusion network model that can cope with multi-tasks, and enhance the transparency and interpretability of the model on specific tasks.

## Figures and Tables

**Figure 1 sensors-24-04065-f001:**
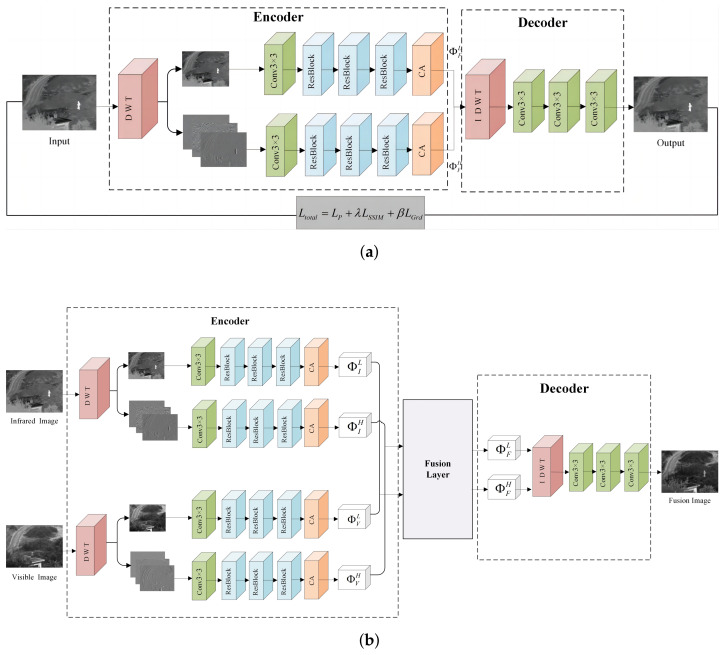
(**a**) The framework of the training process. (**b**) Architecture of the proposed infrared and visible image fusion network.

**Figure 2 sensors-24-04065-f002:**
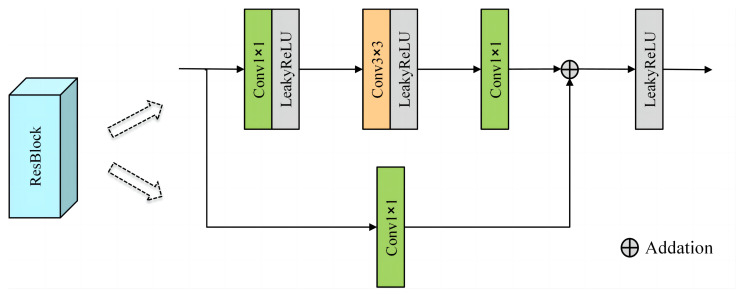
Residual blocks.

**Figure 3 sensors-24-04065-f003:**
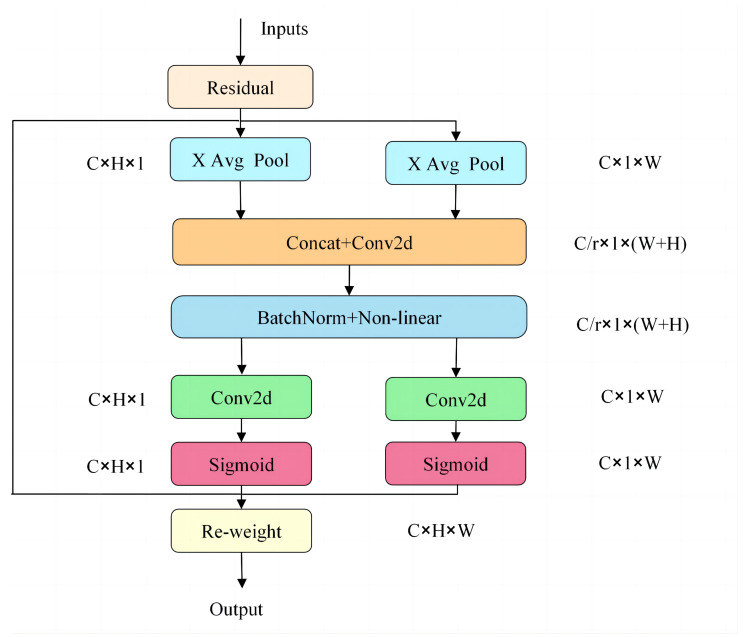
Coordinate attention.

**Figure 4 sensors-24-04065-f004:**
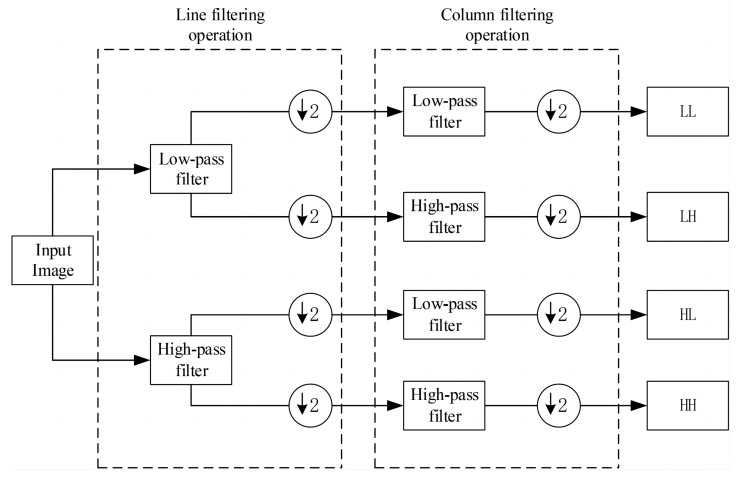
Two-dimensional discrete wavelet transform decomposition process. DWT uses low-pass and high-pass filters to compute the wavelet coefficients, where $L=\frac{1}{\sqrt{2}}{\left[1,1\right]}^{T},H=\frac{1}{\sqrt{2}}{\left[1,1\right]}^{T}$.

**Figure 5 sensors-24-04065-f005:**
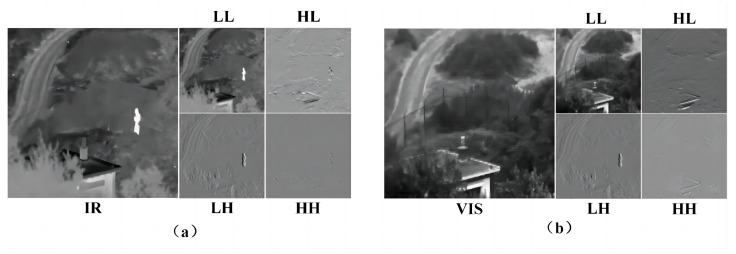
DWT-based image decomposition: (**a**) feature visualization of IR image and (**b**) feature visualization of visible image. IR is the IR input image, and VIS is the visible input image, respectively, which correspond to the four sub-band images LL, LH, HL, and HH after DWT decomposition.

**Figure 6 sensors-24-04065-f006:**
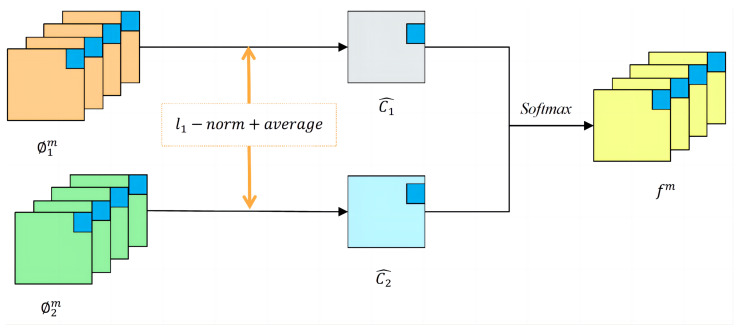
$l_{1}-norm$ fusion strategy.

**Figure 7 sensors-24-04065-f007:**
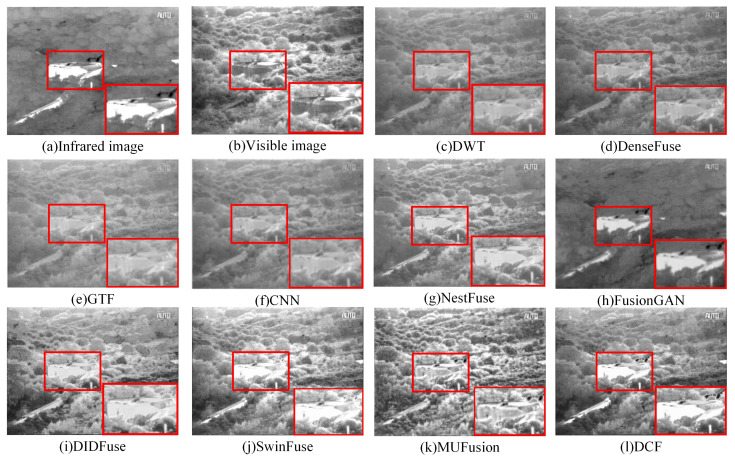
Fusion results of DCF with nine other fusion methods in 2-men-in-front-of-house. For a clear comparison, we selected a prominent region (i.e., the red box) in each image and zoomed in to facilitate a detailed demonstration of the differences in fusion effects.

**Figure 8 sensors-24-04065-f008:**
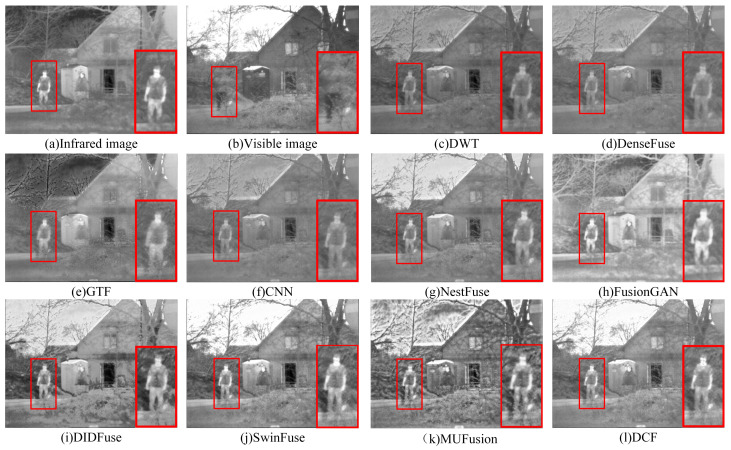
Fusion results of DCF with nine other fusion methods in the image soldier-behind-smoke-1. For a clear comparison, we selected a prominent region (i.e., the red box) in each image and zoomed in to facilitate a detailed demonstration of the differences in fusion effects.

**Figure 9 sensors-24-04065-f009:**
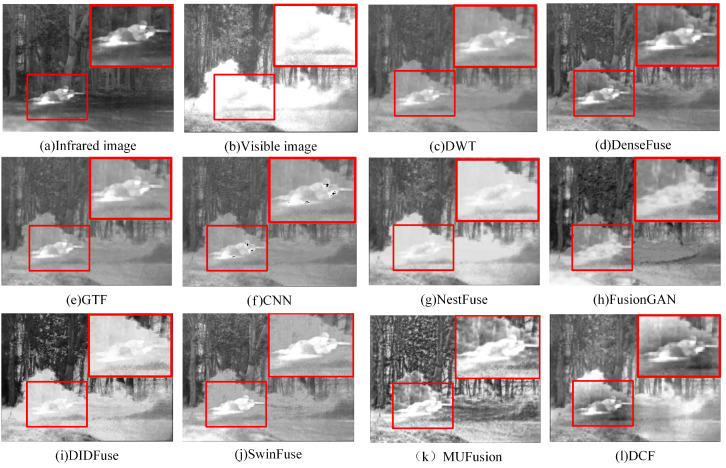
Fusion results of DCF with other nine fusion methods in image bunker. For a clear comparison, we selected a prominent region (i.e., the red box) in each image and zoomed in to facilitate a detailed demonstration of the differences in fusion effects.

**Figure 10 sensors-24-04065-f010:**
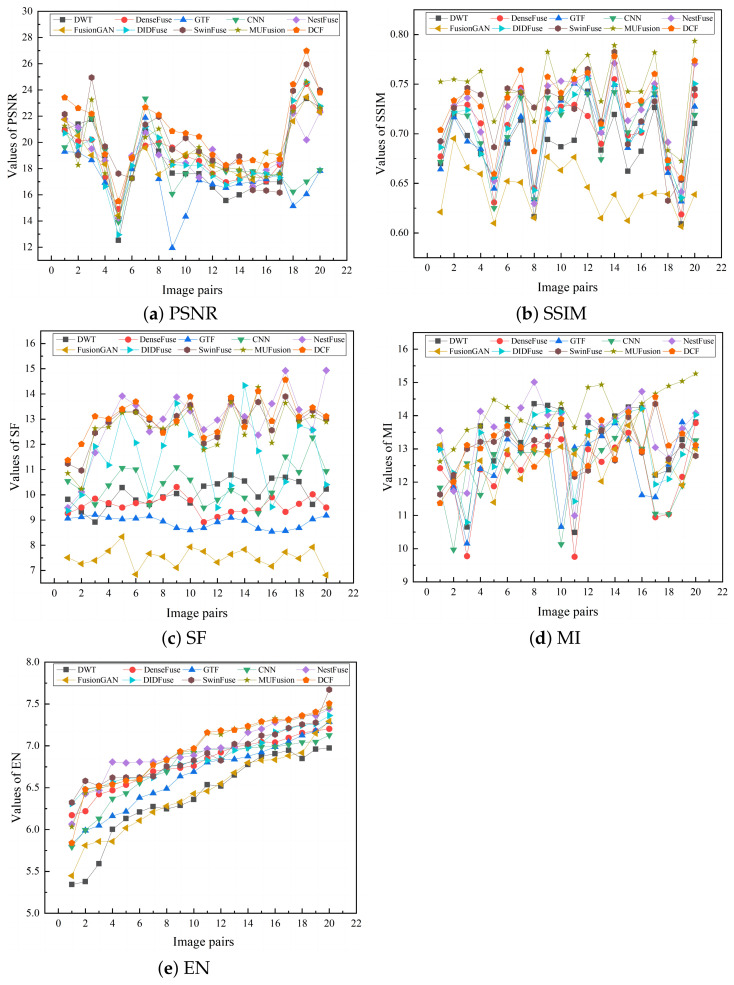
Line plots of the scores of DCF and the remaining 9 state-of-the-art fusion methods on different objective metrics on 20 pairs of images: (**a**) PSNR, (**b**) SSIM, (**c**) SF, (**d**) MI, and (**e**) EN.

**Figure 11 sensors-24-04065-f011:**
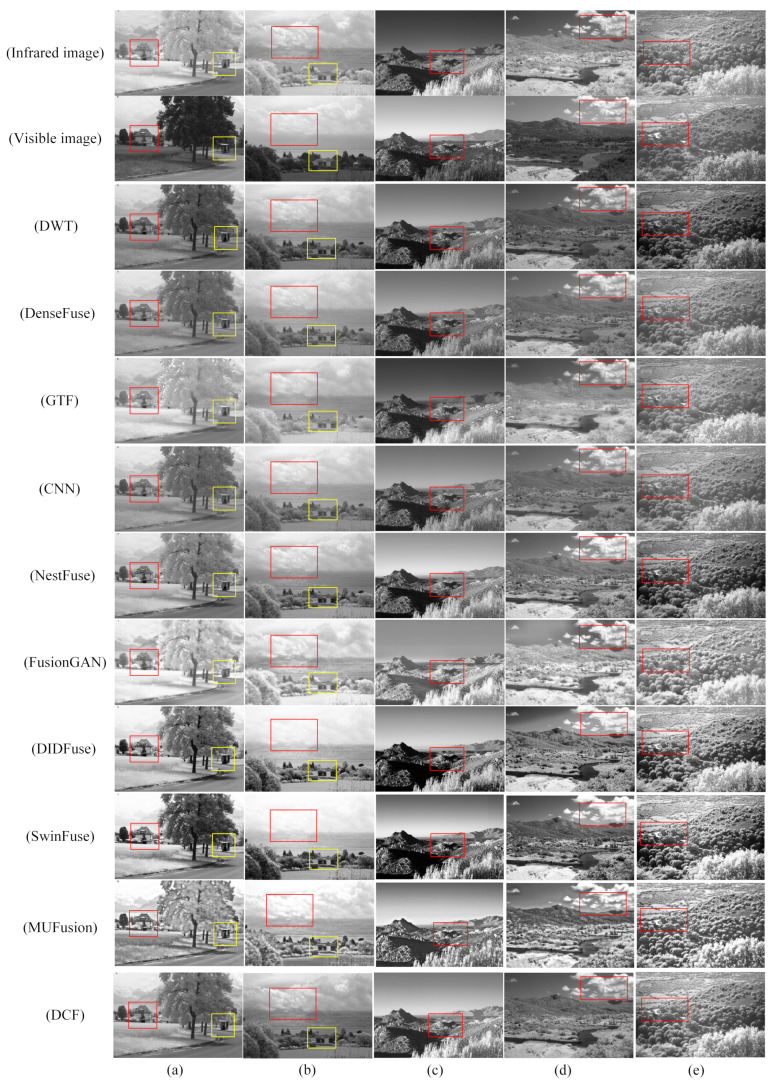
Qualitative comparison of DCF and the remaining nine advanced fusion methods on the NIR dataset. (**a**–**e**) are the images selected in the NIR dataset. For the sake of a clear comparison, we selected the salient regions (i.e., red and yellow boxes) in each image in order to show the difference in fusion effects in detail. Each row from top to bottom is as follows: IR image. Visible image. DWT. DenseFuse. GTF. CNN. Nestfuse. FusionGAN. DIDFuse. SwinFuse. MUFusion. DCF.

**Figure 12 sensors-24-04065-f012:**
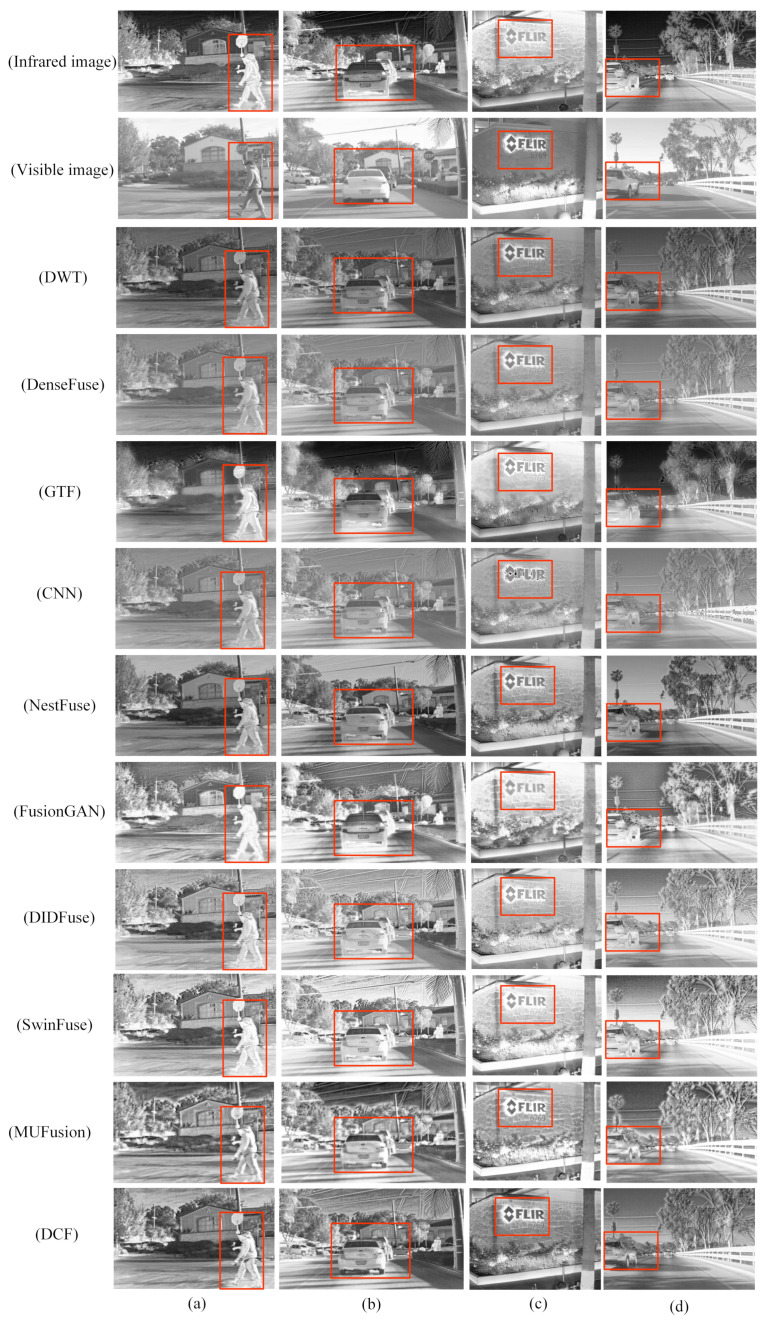
Qualitative comparison of DCF and the remaining nine advanced fusion methods on the RoadScene dataset. (**a**–**d**) are images selected on the RoadScene dataset. For a clear comparison, we selected a prominent region (i.e., the red box) in each image to facilitate a detailed demonstration of the differences in fusion effects. Each row from top to bottom is as follows: IR image. Visible image. DWT. DenseFuse. GTF. CNN. Nestfuse. FusionGAN. DIDFuse. SwinFuse. MUFusion. DCF.

**Figure 13 sensors-24-04065-f013:**
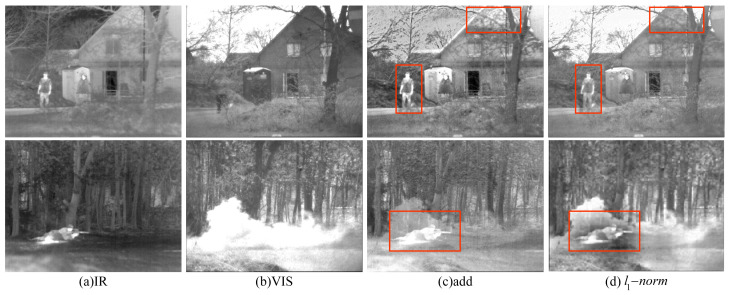
Fusion results for different fusion strategies, where, in order, (**a**) IR image, (**b**) visible image, (**c**) additive strategy, and (**d**) $l_{1}-norm$. In order to clearly compare the experimental results, significant regions are marked with red boxes in this section so that the differences in fusion effects can be observed.

**Figure 14 sensors-24-04065-f014:**
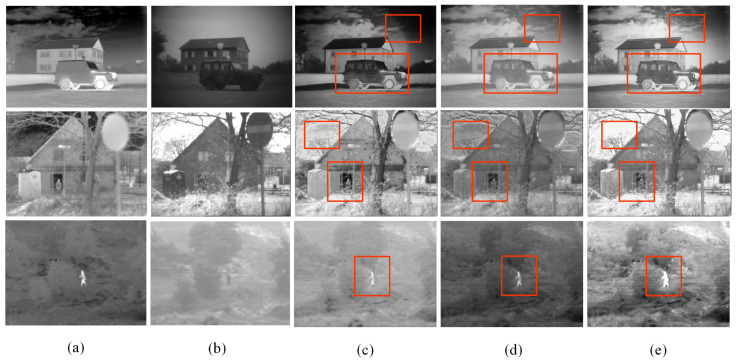
Fusion result plots of different ablation models, where, in order, (**a**) IR image, (**b**) visible image, (**c**) DCF-ND, (**d**) DCF-NC, and (**e**) DCF.

**Figure 15 sensors-24-04065-f015:**
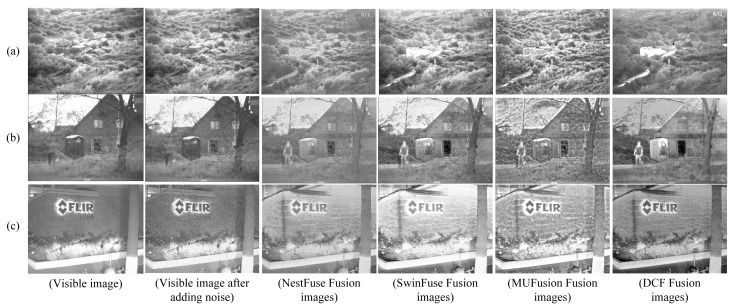
Comparison of fusion results after adding noise. (**a**–**c**) are three different images. From left to right, each row is as follows: visible-light image, visible-light image after adding noise, fusion image of NestFuse, fusion image of SwinFuse, fusion image of MUFusion, and fusion image of DCF.

**Table 1 sensors-24-04065-t001:** Mean quantitative values of PSNR, SSIM, SF, MI, and EN objective metrics for the fusion results of different fusion algorithms on the TNO dataset. Corresponding optimal values are shown in red font, and sub-optimal values are shown in blue font.

Methods	PSNR	SSIM	SF	MI	EN
DWT	17.5967	0.6978	10.3852	12.9652	6.6356
DenseFuse	18.6826	0.7008	9.3236	13.2014	6.7651
GTF	16.8536	0.6689	9.0762	13.3569	6.7414
CNN	17.2893	0.6952	11.4633	13.1536	6.7509
NestFuse	18.7692	0.7312	12.5629	13.8563	6.7864
FusionGAN	18.7138	0.6315	7.5592	12.6526	6.4523
DIDFuse	18.4856	0.7159	12.3657	13.1639	6.7608
SwinFuse	20.2528	0.7408	12.8635	13.6335	6.7849
MUFusion	19.1896	0.7736	12.8235	14.8654	6.8263
DCF	20.3948	0.7439	13.1623	13.7502	6.8836

**Table 2 sensors-24-04065-t002:** Mean quantitative values of PSNR, SSIM, SF, MI, and EN objective metrics for the fusion results of different fusion algorithms on the TNO dataset. Corresponding optimal values are shown in red font, and sub-optimal values are shown in blue font.

Methods	PSNR	SSIM	SF	MI	EN
DWT	18.4916	1.2775	18.2348	13.2539	6.8637
DenseFuse	19.9635	1.3823	19.6792	14.4587	7.2187
GTF	18.6393	1.2689	19.1652	14.3788	6.9631
CNN	18.8623	1.2397	20.4506	13.7812	6.9729
NestFuse	20.9345	1.4626	22.8376	14.6938	7.2563
FusionGAN	18.9634	1.2581	15.4589	13.8469	7.0056
DIDFuse	19.7553	1.3941	22.9251	14.1459	7.1068
SwinFuse	21.0523	1.5632	22.3515	13.8916	7.2946
MUFusion	20.1694	1.3649	23.0764	14.0235	7.3944
DCF	21.6876	1.4915	23.4386	14.4089	7.4614

**Table 3 sensors-24-04065-t003:** Mean quantitative values of PSNR, SSIM, SF, MI, and EN objective metrics for the fusion results of different fusion algorithms on the TNO dataset. Corresponding optimal values are shown in red font, and sub-optimal values are shown in blue font.

Methods	PSNR	SSIM	SF	MI	EN
DWT	17.6827	0.8869	15.6345	13.1608	6.8123
DenseFuse	19.4508	1.0821	17.6952	13.8147	7.0439
GTF	17.2318	0.8363	15.9154	13.7914	6.8146
CNN	17.9361	0.8507	16.9855	13.4637	6.8413
NestFuse	20.3067	1.1654	18.2218	14.2935	7.1639
FusionGAN	18.8296	0.8604	11.3492	13.6463	6.7924
DIDFuse	19.2123	0.9861	17.8477	13.9639	6.9116
Swinfuse	19.8862	1.2594	18.1236	14.0828	7.1129
MUFusion	17.2638	1.0362	18.2418	13.4198	7.1671
DCF	21.2015	1.2361	18.5688	14.1369	7.2948

**Table 4 sensors-24-04065-t004:** Mean quantitative values of PSNR, SSIM, SF, MI, and EN objective metrics for the fusion results of different fusion algorithms on the TNO dataset. Corresponding optimal values are shown in red font, and sub-optimal values are shown in blue font.

Methods	PSNR	SSIM	SF	MI	EN
add	15.4923	0.6248	14.8176	12.8796	6.7413
$l_{1}-norm$	15.4817	0.6379	15.8107	13.7502	6.8836

**Table 5 sensors-24-04065-t005:** Mean quantitative values of PSNR, SSIM, SF, MI, and EN objective metrics for the fusion results of different fusion algorithms on the TNO dataset. Corresponding optimal values are shown in red font, and sub-optimal values are shown in blue font.

Methods	PSNR	SSIM	SF	MI	EN
DCF-ND	15.3245	0.6226	13.2612	12.7867	6.5571
DCF-NC	15.2364	0.6185	14.2394	12.7236	6.5418
DCF	15.4817	0.6379	15.8107	13.7502	6.8836

**Table 6 sensors-24-04065-t006:** Comparison of the computational efficiency (in seconds) of 10 different fusion algorithms on three datasets (red indicates the optimal value; blue indicates the sub-optimal value).

Methods	TNO	NIR	RoadScene
DWT	10.5152	8.8926	7.4854
DenseFuse	0.3246	0.4625	0.1756
GTF	2.6562	2.2632	1.9625
CNN	3.4512	2.9862	3.3494
NestFuse	0.1938	0.0856	0.1465
FusionGAN	0.5265	0.4593	0.4804
DIDFuse	0.2761	0.2462	0.1729
SwinFuse	0.3361	0.2256	0.2195
MUFusion	0.2963	0.1928	0.2362
DCF	0.2632	0.1697	0.1665

## Data Availability

Data are contained within the article.

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
