# Peer review of "DCFNet: Infrared and Visible Image Fusion Network Based on Discrete Wavelet Transform and Convolutional Neural Network"

_sensors, 2024, doi:10.3390/s24134065_

Round 1

Reviewer 1 Report

Comments and Suggestions for Authors

1) The innovation and key points of the abstract and contributions are not highlighted enough. That is how the DWT and IDWT components help to solve the problem of missing detail information, blurring of significant target information and poor fusion visual effect. 

2) Quite a few papers related to image fusion based on DWT are missing from references. For example:

- Singh S , Vig R , Kaur G. Medical Fusion Framework Using Discrete Fractional Wavelets and Non Sub-Sampled Directional Filter Banks[J].IET Image Processing, 2019, 14(4).DOI:10.1049/iet-ipr.2019.0948.

- Paul Hill et al., "Perceptual Image Fusion Using Wavelets," IEEE TRANSACTIONS ON IMAGE PROCESSING, VOL. 26, NO. 3, MARCH 2017

- Haonan Su et al.,"Multi-Spectral Fusion and Denoising of RGB and NIR Images Using Multi-Scale Wavelet Analysis," ICPR, 2018.

And also some similar methods have been already considered for near-infrared fusion, flash image fusion, and noisy/blurred image fusion. Therefore, authors should add some related papers to the references and summarize various types of image fusion models with a table.

3) Most of the image fusion methods [22,26,27,29,34,35,36] used in this paper are out of date. Author should compare your method with the recently-introduced state-of-the-art methods. 

4) Partial references information are incomplete, such as: 21,34. 

5) Why use DWT and IDWT module to decompose and reconstruct both the infrared and visible image? It would cause the edge blurring of target.

6) Suggest supplementing the simulation results of the proposed algorithm in noisy image fusion.

7) Perform a syntax, style and grammar check throughout the manuscript to improve its readability.

Comments on the Quality of English Language

 Perform a syntax, style and grammar check throughout the manuscript to improve its readability.

Reviewer 2 Report

Comments and Suggestions for Authors

Dear Authors,

I have some comments and requests for changes:

  1. At the end of the Introduction section, please include a description of the paper's organization.
  2. On Page 6 of 25, Line 205, where you mention "except the middle convolutional layer which uses 3×3 convolutional kernels," please provide the reasoning for choosing 3x3 for the middle convolutional layer.
  3. On Page 6 of 25, Line 216, specify the type of padding used in the system.
  4. Define the variable k in Equations 7, 8, and 9.
  5. Clarify the difference between Equations 20 and 21.
  6. On Page 19 of 25, Line 467, for PSNR, SSIM, SF, and MI, it seems it should be EN instead of MI.
  7. Include some numerical results in the Conclusion section.
Comments on the Quality of English Language

Some typos and punctuation have to be rechecked.

Reviewer 3 Report

Comments and Suggestions for Authors

This article utilizes the fusion of visible light and infrared images to improve image clarity and visualization effects. This study has certain engineering significance. The author presents two contributions: the first is to use discrete wavelet transform to decompose the global features of low frequencies and the detailed features of high frequencies in images, optimizing the processing performance of the network; The second one utilizes residual modules and attention mechanisms.

This judge believes that this article still needs to be revised to better highlight its innovation:

1. At present, these public datasets, including TNO, NIR, and RoadScene, have been widely used by many people, and the algorithm proposed by the author is actually competing with numerous algorithms. So the characteristics of this algorithm need to be more clear, such as computational complexity, computational time, and the need for computer resources, all of which require necessary analysis and comparison to demonstrate the advantages of this algorithm;

2On the basis of adding steps such as discrete wavelet transform, the author has achieved a slight improvement compared to traditional algorithms. The author needs to introduce the shortcomings of this algorithm? Or rather, at what cost have these improvements in performance been achieved?

Comments on the Quality of English Language

nothing
